

# The silicon stable isotope distribution along the GEOVIDE section of the North Atlantic Ocean

Jill N. Sutton[1], Gregory F. de Souza[2], Maribel I. García-Ibáñez[3,4] and Christina L. De La Rocha[1,5]

[1]Université de Brest, UMR 6539 CNRS/UBO/IRD/Ifremer, LEMAR, IUEM, 29280, Plouzané, France
[2]ETH Zurich, Institute of Geochemistry and Petrology, Clausiusstrasse 25, 8092 Zurich, Switzerland
[3]Uni Research Climate, Bjerknes Centre for Climate Research, Bergen 5008, Norway
[4]Instituto de Investigaciones Marinas, IIM-CSIC, Eduardo Cabello 6, 36208 Vigo, Spain
[5]Currently without affiliation

*Correspondence to*: Jill N. Sutton (jill.sutton@univ-brest.fr)

**Abstract.** The stable isotope composition of dissolved silicon in seawater ($\delta^{30}Si_{DSi}$) was examined at 10 stations along the GEOVIDE section, spanning the North Atlantic Ocean (40°N-60°N) and Labrador Sea. Near-surface water $\delta^{30}Si_{DSi}$ could not be evaluated due to the very low dissolved silicon (DSi) concentrations (< 5 μM). However, variations in $\delta^{30}Si_{DSi}$ below 500 m were closely tied to the distribution of water masses. Higher $\delta^{30}Si_{DSi}$ values are associated with intermediate and deep water masses of northern Atlantic or Arctic Ocean origin, whilst lower $\delta^{30}Si_{DSi}$ values are associated with DSi-rich waters sourced
ultimately from the Southern Ocean. Correspondingly, the lowest $\delta^{30}Si_{DSi}$ values were observed in the deep and abyssal eastern North Atlantic, where dense southern-sourced waters dominate. The extent to which the spreading of water masses influences the $\delta^{30}Si_{DSi}$ distribution is marked clearly by Labrador Sea Water (LSW), whose high $\delta^{30}Si_{DSi}$ signature is visible not only within its region of formation within the Labrador and Irminger Seas, but also throughout the mid-depth western and eastern North Atlantic Ocean. Both $\delta^{30}Si_{DSi}$ and hydrographic parameters document the circulation of LSW into the eastern North
Atlantic, where it overlies southern-sourced Lower Deep Water. The GEOVIDE $\delta^{30}Si_{DSi}$ distribution thus provides a clear view of the direct interaction between subpolar/polar water masses of northern and southern origin, and allow examination of the extent to which these far-field signals influence the local $\delta^{30}Si_{DSi}$ distribution.

## 1 Introduction

Proxies of nutrient utilisation, such as the silicon stable isotopic composition ($\delta^{30}Si$) of diatom silica, provide a means of
reconstructing the past behaviour of marine nutrient cycles, giving insight into the strength of the biological pump in the past, and its influence over atmospheric concentrations of $CO_2$. However, diatom silica $\delta^{30}Si$ does not depend solely on the degree of utilisation of dissolved silicon (DSi) at the ocean's surface, but also on the $\delta^{30}Si$ value of its source DSi. Since the $\delta^{30}Si$ of DSi ($\delta^{30}Si_{DSi}$) at any given location in the ocean results from the combined effects of biological uptake of dissolved silicon, dissolution of sinking biogenic silica, and meso- to macro-scale features of ocean circulation, successfully reconstructing past



silica cycling from the variations in $\delta^{30}$Si of diatoms accumulating in sediments requires a reasonable understanding of the processes that control the $\delta^{30}$Si$_{DSi}$ distribution.

Significant progress has been made in this regard by fifteen years' worth of work in the Southern Ocean (Varela et al. 2004;

Cardinal et al., 2005; De La Rocha et al., 2011; Fripiat et al., 2011), in the North, Equatorial and South Pacific (De La Rocha et al., 2000; Reynolds et al., 2006; Beucher et al., 2008; 2011; de Souza et al., 2012a), and recently in the Arctic Ocean (Varela et al., 2016), in conjunction with various models (De La Rocha and Bickle, 2005; Reynolds, 2009; Coffineau et al., 2014), not the least of which are global circulation models (Wischmeyer et al., 2003; de Souza et al., 2014; 2015; Holzer and Brzezinski, 2015). It is now widely understood that fractionation of silicon isotopes during uptake and biomineralization of silica in surface

waters increasingly elevates the $\delta^{30}$Si$_{DSi}$ in surface waters (De La Rocha et al., 1997; Sutton et al., 2013). At the same time, dissolution of biogenic silica exported to deeper layers works to enrich them in dissolved silicon of lower $\delta^{30}$Si$_{DSi}$ (Demarest et al., 2009; de Souza et al., 2014; Wetzel et al., 2014). For many deep waters of the ocean, the mixing between water masses of vastly different origin (and thus different $\delta^{30}$Si$_{DSi}$) as they circulate through the ocean basins plays a much greater role than the dissolution of sinking biogenic silica in setting geographic patterns in deep ocean $\delta^{30}$Si$_{DSi}$ (de Souza et al., 2014).

This is particularly true of the deep Atlantic Ocean, which displays a notable north-south gradient in both the concentrations of dissolved silicon (from < 10 µM in the North Atlantic to > 125 µM in the South Atlantic) and its $\delta^{30}$Si$_{DSi}$ (from roughly +1.9 ‰ in the North Atlantic down to +1.2 ‰ in the South Atlantic) (de Souza et al., 2012b; Brzezinski and Jones, 2015). Based on modeling results (de Souza et al., 2014), the bulk of the change in $\delta^{30}$Si$_{DSi}$ occurs mainly in the northern North

Atlantic for reasons that are unique to this area of the ocean. The Labrador Sea, located between Greenland and the North American continent, is a key site contributing to the formation of North Atlantic Deep Water (NADW). The surface waters that cool in this region and sink to form Labrador Sea Water (LSW), an important component of NADW, are nutrient-poor. This means that their DSi concentration is markedly low and its $\delta^{30}$Si$_{DSi}$ is notably high, characteristics that are imparted to the deep water mass during formation. Its low DSi concentration makes the $\delta^{30}$Si$_{DSi}$ of deep water in this area very susceptible to

change via the addition of dissolved silicon by mixing or dissolution of opal. Unfortunately, only one depth profile of $\delta^{30}$Si$_{DSi}$ is currently available for the entire Labrador Sea DSi (de Souza et al., 2012b). Even without taking into account its dynamic nature, the North Atlantic bears better mapping of the $\delta^{30}$Si$_{DSi}$ of its waters.

Recently, we began rectifying the situation via an internationally-funded **GEO**TRACES campaign that was carried out along

the **O**VIDE section of the North Atlantic Ocean and Labrador Sea (GEOVIDE). We use the $\delta^{30}$Si$_{DSi}$ distribution to constrain the processes that influence the distribution and cycling of silica in the North Atlantic Ocean and Labrador Sea.



## 2 Materials & Methods

### 2.1 Sample collection and processing

Samples were collected aboard the R/V *Pourquoi Pas?* during GEOVIDE. The cruise began on May 15, 2014 in Lisbon, Portugal, headed north towards Greenland, and then traversed south-west to St. John's, Canada, arriving on June 30, 2014

(Fig. 1).

Seawater samples for $\delta^{30}Si_{DSi}$ analysis were collected using Niskin bottles attached to a standard rosette conductivity-temperature-depth (CTD) unit from 10 locations in the North Atlantic Ocean and Labrador Sea (Fig. 1 and Table 1). Samples were filtered through polycarbonate 0.45 μm filters (Millipore) and stored in acid-cleaned, low-density polyethylene bottles at

room temperature.

The DSi concentration of all samples collected (n = 56) was determined via molybdate blue spectrophotometry (Strickland and Parsons, 1972). For the measurement of $\delta^{30}Si_{DSi}$, DSi was extracted from the seawater by precipitating it as trimethylamine silicomolybdate, which was subsequently combusted to form $SiO_2$ (De La Rocha et al., 1996). This $SiO_2$ was dissolved at

room temperature in polypropylene microcentrifuge tubes (1.5 mL) containing 23 M HF (Suprapur) to yield a final solution concentration of 0.23 M Si (e.g. 4 μmol of $SiO_2$ dissolved in 17.4 μL of HF).

### 2.2 Purification

Every sample was further purified using ion exchange chromatography following Engström et al. (2006). Briefly, 17.4 μL of a 0.23 M Si solution was diluted in 7.7 mL of Ultra Hiqh Quality water (UHQ $H_2O$; 18.2 MΩ-cm; Millipore Direct-Q) and

loaded onto columns containing AG 1-X8 resin (BioRad) that had been preconditioned with 2 M NaOH. The sample matrix was eluted using 95 mM HCl + 23 mM HF followed by the elution of the purified Si using 0.14 M HNO3 + 5.6 mM HF. All acids were Suprapur (Merck) and were diluted with UHQ $H_2O$.

### 2.3 Isotopic measurements

Silicon stable isotope composition ($\delta^{30}Si$) of the purified samples was measured using standard-sample bracketing combined

with external normalisation by doping the samples with magnesium (Mg) (e.g., Cardinal et al., 2003) on a Thermo Scientific *Neptune* multi-collector inductively-coupled plasma mass spectrometer (MC–ICP-MS) at the Unité Géosciences Marines (Ifremer, Plouzané). Prior to the isotopic analysis, the purified samples were diluted with 0.16 M $HNO_3$ (1 % $HNO_3$) to 1 ppm Si, yielding a roughly 12 V signal on mass 28 at medium resolution (see Table 2 for additional information on operating conditions). All samples and standards (NBS28 and a 99.995 % pure silica sand (Alfa Aesar) used as a working standard) were

passed through column chemistry and matrix-matched to give the same signal strength (within 10 %) and to contain the same amount of HF (generally 1 mM). Magnesium (1000 ppm, NIST SRM) was added to the samples and standards just prior to



measurement at a final concentration of 0.1 ppm (Cardinal et al., 2003; Abraham et al., 2008). Si solutions were introduced into the plasma via an Apex desolvating system equipped with a PFA nebulizer (uptake rate = 100 μL min$^{-1}$) without additional gas.

For each measurement, beam intensities at masses 25 and 26 (Mg), and 28, 29, and 30 (Si) were monitored in dynamic mode (i.e. switching between Si and Mg masses) for one block of 25 cycles of 8-second integrations. Five minutes of rinse with 2 % HNO$_3$ followed each sample and each standard solution. Solutions were analyzed in medium resolution mode (m/Δm > 6000). Using a standard-sample-standard bracketing technique, $\delta^{30}Si$ values for the samples were expressed as follows:

$\delta^{30}Si$ (‰) = [($^{30}Si/^{28}Si$)/($^{30}Si/^{28}Si)_{standard}$)-1] × 1000               (1)

The Si isotope ratios in Eqn. 1 above ($^{30}Si/^{28}Si$ and $^{29}Si/^{28}Si$) were corrected for mass bias by external normalisation using Mg doping. For example, the corrected $^{30}Si$ to $^{28}Si$ ratio ($^{30}Si/^{28}Si)_{corr}$ is:

$(^{30}Si/^{28}Si)_{corr} = (^{30}Si/^{28}Si)_{meas}$ x $(^{30}Si_{AM}/^{28}Si_{AM})^{\varepsilon Mg}$              (2)

Where $(^{30}Si/^{28}Si)_{meas}$ is the measured ratio, $^{30}Si_{AM}$ and $^{28}Si_{AM}$ are the atomic masses of $^{30}Si$ and $^{28}Si$. $\varepsilon_{Mg}$ is calculated from the beam intensities on masses 25 and 26 :

$\varepsilon_{Mg}$ = ln [$(^{25}Mg_A/^{26}Mg_A)/(^{5}Mg/^{26}Mg)_{meas}$]/[ $^{25}Mg_{AM}/^{26}Mg_{AM}$]            (3)

where $^{25}Mg_A/^{26}Mg_A$ is the expected ratio of the natural abundances of the isotopes, $(^{25}Mg/^{26}Mg)_{meas}$ is the measured ratio, and $^{25}Mg_{AM}$ and $^{26}Mg_{AM}$ are the atomic masses of $^{25}Mg$ and $^{26}Mg$ respectively. Each measurement of a sample fell between two measurements of the standard, and each sample was measured three times. This total of three sample measurements and five 25   standard measurements was repeated 2-3 times in each mass spectrometry session and used to calculate one replicate value of $\delta^{30}Si$ and $\delta^{29}Si$. As discussed below, full chemistry replicates were routine for each sample (see Table S2).

Interference-free measurement was ensured by checking that $\delta^{29}Si$ and $\delta^{30}Si$ for all samples was consistent with the mass dependent fractionation line (Fig. 2). The signal was optimized to reduce the $^{14}N^{16}O$ interference on m/z 30 to below 0.5 % of 30   the $^{30}Si$ peak. Measurements were performed on the low-mass side of the peak where interference is minimal. Blanks were maintained below 1 % of the main signal and were subtracted for each sample and standard. Long-term reproducibility and accuracy on $\delta^{30}Si$ values of the analytical procedure were assessed using the standard deviation of 54 analyses of NBS28 and 29 analyses of a secondary reference standard (Silicon (IV) oxide, Alfa Aesar) generated over 6 years (±0.10 ‰, 2σ).



Reproducibility of the full chemical and analytical procedure was estimated using at least one replicate of each sample (chemical preparation plus isotopic measurements) and average reproducibility on replicate $\delta^{30}Si$ was ±0.10 ‰ (2σ). Measurements of Big Batch (n = 3) produced an average value of -10.48 ± 0.34 (2σ), well within the range of intercalibration values reported by Reynolds et al. (2007). Measurement of the US GEOTRACES intercalibration reference seawater standard

from the Aloha Station (1000 m) gave a $\delta^{30}Si_{DSi}$ value of +1.16 ± 0.16 ‰ (2σ, n = 3), within the range of intercalibration values (1.24 ± 0.20 ‰; Grasse et al., 2017). Measurement of the Canada/GEOVIDE GEOTRACES intercalibration samples, where duplicate samples at 3 depths and 2 stations were analysed by two different laboratories (Ifremer Plouzané and ETH Zurich), conforming to the GEOTRACES intercalibration protocol for a cruise without a cross-over station, gave similar $\delta^{30}Si_{DSi}$ values (see Table 3). Note that ETH Zurich uses a different purification method (cation exchange resin, see de Souza

et al., 2012b) and MC-ICPMS instrument (Nu Plasma 1700) than Ifremer Plouzané (described in section 2.2). The methods for each laboratory that participated in Canada/GEOVIDE GEOTRACES intercalibration study are also presented in the US GEOTRACES intercalibration study (Grasse et al., 2017).

**2.2 Optimum Multiparameter Analysis to determine the water mass structure in the North Atlantic Ocean**

In order to accurately examine the relationship between the distribution of $\delta^{30}Si_{DSi}$ and water masses, the results of the

Optimum Multiparameter (OMP) analysis of García-Ibáñez et al. (2017; this issue) were used to identify the mixture of water masses present within each sample and their contribution to the DSi budget.

The upper layers of the GEOVIDE section were represented by the North Atlantic Central Waters (NACW), transported by the North Atlantic Current (NAC; Pollard et al., 1996), and Subpolar Mode Waters (SPMW), the end-product of the

transformation of NACW through air-sea interaction (McCartney and Talley, 1982; Tsuchiya et al., 1992). To account for the change in the temperature of SPMW along the path of the NAC as the result of air-sea interaction, two SPMWs were differentiated: IcSPMW (Iceland-Subpolar Mode Water) and IrSPMW (Irminger-Subpolar Mode Water). The intermediate layers of the section were represented by LSW, Mediterranean Water (MW), Subarctic Intermediate Water (SAIW), and Polar Intermediate Water (PIW). LSW is the last stage of the transformation of SPMWs and forms in the Labrador and Irminger

Seas (e.g., Pickart et al., 2003; de Jong and de Steur, 2016; Fröb et al., 2016). MW enters the North Atlantic from the Mediterranean Sea through the Strait of Gibraltar (Ambar and Howe, 1979; Baringer and Price, 1997). SAIW originates in the Labrador Current by mixing of the NAC waters with LSW (Iselin, 1936; Arhan, 1990; Read, 2000). The deep layers of the section were represented by Denmark Strait Overflow Water (DSOW), Iceland–Scotland Overflow Water (ISOW), North East Atlantic Deep Water (NEADW) and Lower Deep Water (LDW). Overflow waters (DSOW and ISOW) form after the deep

waters of the Nordic Seas flow over the Greenland–Iceland–Scotland sills and entrain Atlantic waters (van Aken and de Boer, 1995; Read, 2000; Dickson et al., 2002; Fogelqvist et al., 2003; Yashayaev and Dickson, 2008). NEADW is formed as a result of entrainment events that occur along the journey of ISOW through the Iceland Basin (van Aken, 2000). NEADW recirculates




in the West European Basin and mixes with the surrounding waters, including the Antarctic Bottom Water (AABW) (van Aken and Becker, 1996), resulting in the formation of LDW.

## 3 Results

### 3.1 Water column profiles of DSi

5    Dissolved silica concentrations below 500 m along the GEOVIDE section ranged from 7 to 47 μM (Fig. 3a). The stations located to the east of the Mid-Atlantic Ridge (MAR; STN 01, STN 13, STN 21, STN 26, STN 32) show DSi increasing in concentration from < 10 μM to 20-50 μM below about 2000 m (Fig. 3a). Stations located to the west of the MAR (STN 44, STN 60, STN 64, STN 69, and STN 70) show only slight increases in DSi concentration with depth, with most of the values falling between 9-12 μM (Fig. 3a). This difference relates to the distribution of water masses in the northern North Atlantic,

10    with the predominance of the most egregiously Si-poor northern-sourced water masses (LSW, ISOW/NEADW, DSOW) predominating in the western Atlantic while abyssal layers in the eastern Atlantic have had more of a contribution from Si-rich southern-sourced waters (LDW) (see Section 4.1). A clear pattern in the DSi concentration throughout the water column is that the eastern profiles exhibit a more typically "nutrient-like" profile than in the western profiles (Supplementary Table S1; Fig. 3a).

### 15   3.2 Water column profiles of $\delta^{30}Si_{DSi}$

All GEOVIDE water column profiles have relatively high $\delta^{30}Si_{DSi}$ (+1.5 to +3 ‰) between 500-1000 m, and show a trend towards lower $\delta^{30}Si_{DSi}$ values with depth (although none significantly lower than +1 ‰) (Fig. 3b). Strikingly, the lowest values of $\delta^{30}Si_{DSi}$ occur at stations nearer to the Iberian margin (e.g., STN 01, STN 13), while highest values tend to occur at the upper depths of the profiles (500-1000 m) at stations north of 50°N and west of 20°E (Stations 26, 32, 44, 60, 64, 69, 77; Fig. 3b),

20    mirroring the differences in DSi between these locations (Fig. 3a, Supplementary Tables S1 and S2). Profiles of $\delta^{30}Si_{DSi}$ in the Labrador Sea (stations 64, 69, 77) show high values (above +1.5 ‰) extending below 2000 m water depth. These high values reach the bottom within the central Labrador Sea (STN 69). Elevated $\delta^{30}Si_{DSi}$ values at depths between 1000-2000 m can be found at all stations in the Irminger Basin and extend eastwards into the western portion of the West European Basin (up to STN 26), but the mid-depths of the far eastern Atlantic are marked by lower $\delta^{30}Si_{DSi}$ values around +1 ‰.

## 25   4 Discussion

### 4.1 North Atlantic $\delta^{30}Si_{DSi}$ systematics in a basin-wide context

The deep Atlantic Ocean below 1000 m exhibits a wide variation in DSi concentrations, ranging from ~10 μM in the mid-depths of the subpolar North Atlantic Ocean to ~120 μM in the abyssal southern Atlantic. Since at least the work of Broecker





and Takahashi (1980), it has been known that this variation is primarily brought about by the quasi-conservative mixing of DSi between Si-rich abyssal waters derived from the Southern Ocean and Si-poor waters of North Atlantic origin. The analysis of Sarmiento et al. (2007), which takes the effects of water mass mixing into account, has shown that the effect of opal dissolution on deep Atlantic DSi is resolvable, but plays a near-negligible role in controlling the deep DSi distribution.

The first systematic study of the Atlantic $\delta^{30}Si_{DSi}$ distribution (de Souza et al., 2012b) showed that the quasi-conservative behaviour of DSi is clearly reflected by the $\delta^{30}Si_{DSi}$ of the deep Atlantic Ocean. Surveying deep water over a wide range of latitudes within the Atlantic Ocean, they found that values of $\delta^{30}Si_{DSi}$ vary coherently from high values in the Si-poor waters that contribute to NADW to low values in the Si-rich Southern Ocean deep waters (Fig. 4). The more recent work of Brzezinski

and Jones (2015) found identical behaviour within Atlantic deep waters along a near-zonal transect across the subtropical North Atlantic (Fig. 4).

Our data agree with the systematics of these two studies, with each data set exhibiting nearly identical linear regressions, except for the value of the y-intercept, which appear to result from a near-constant offset between $\delta^{30}Si_{DSi}$ values measured at

different laboratories (see Fig. 4). Similar offsets were observed and discussed by Brzezinski and Jones (2015), Such offsets of order ±0.2 ‰ have been recognized to exist between seawater $\delta^{30}Si$ data produced in different laboratories; their origin remains unclear, although they may have to do with differences in sample processing and chemical purification. The offset to the data of de Souza et al. (2012b), produced at ETH Zurich, is somewhat surprising given the good agreement in $\delta^{30}Si_{DSi}$ for 6 seawater samples analyzed both at Plouzané and Zurich, but a small offset to lower $\delta^{30}Si_{DSi}$ at Plouzané is consistent with

the offset (0.1 ‰) in these two laboratories' mean $\delta^{30}Si_{DSi}$ values for the seawater reference Aloha-1000 (Grasse et al., 2017). Whilst not ideal for the determination of the absolute $\delta^{30}Si_{DSi}$ value for each basin, the existence of such interlaboratory offsets does not impair our ability to analyze the distribution of $\delta^{30}Si_{DSi}$ along the GEOVIDE transect, with the systematics of our data exhibiting nearly-identical behaviour to previously-published studies (Fig. 4).

Factoring out the offset in absolute $\delta^{30}Si_{DSi}$ values, it is interesting to note that our deep North Atlantic samples exhibit essentially the same $\delta^{30}Si_{DSi}$ range (~0.6 ‰; Fig. 4) as that observed over the entire latitudinal range of the Atlantic Ocean (de Souza et al., 2012b). Our dataset thus indicates that DSi in the North Atlantic Ocean is an important source of isotopic variability in the deep ocean. This is at least partially due to the transport of isotopically heavy DSi to the North Atlantic by northward-flowing Subantarctic Mode Water / Antarctic Intermediate Water (de Souza et a., 2012b; 2015) and its incorporation

into NADW, e.g., during the formation of LSW. However, Brzezinski and Jones (2015) hypothesized that the Arctic Ocean may also play an important role in producing deep North Atlantic $\delta^{30}Si_{DSi}$ variability, via overflows across the Greenland-Iceland-Scotland ridge (i.e., DSOW and ISOW), which dominantly contribute to Lower NADW. It thus remains to be understood how the isotopic compositions of various precursors of NADW contribute to its isotopic signal. In the following,



we discuss our $\delta^{30}Si_{DSi}$ dataset in the context of regional oceanography, in order to study the control of interacting interior water masses on the $\delta^{30}Si_{DSi}$ distribution of the high North Atlantic.

**4.2 Relationship between North Atlantic $\delta^{30}Si_{DSi}$ distribution and water mass structure**

The GEOVIDE section intersects numerous water masses of various origins whose presence is reflected in the distributions of
salinity, dissolved oxygen ($O_2$), and potential vorticity (PV) along the section (Fig. 5a, b, c; dissolved $O_2$ is presented as percent saturation, i.e., $O_2/O_2^{sat} \times 100$, where $O_2^{sat}$ is the saturation $O_2$ concentration). Since $\delta^{30}Si_{DSi}$ was only measured at depths below 500 m, we focus on the intermediate and deep ocean water masses. In discussing the relationship between the $\delta^{30}Si_{DSi}$ distribution and water mass structure, we initially focus on the western- and easternmost sections of the GEOVIDE transect, where water masses are in their most unadulterated form along the transect, prior to discussing their extension into the mid-
Atlantic.

Starting with the westernmost profiles, those of STNs 77, 69, and 64 in the Labrador Sea, we see relatively well-oxygenated waters with low PV (Fig. 5) extending to depths below 2000 m, reflecting the presence of LSW. This water mass, which contributes to NADW, is formed by deep convection in the Labrador and Irminger Seas (e.g., de Jong and de Steur, 2016) and
spreads across the North Atlantic at intermediate to mid-depths (see Fig. 1). Two distinct types of LSW can be distinguished and are indeed visible in our profiles. There is an extremely-low-PV ($< 4\times10^{-12}$ m$^{-1}$ s$^{-1}$) and well-oxygenated ($> 90\,\%$ saturation) pycnostad extending from around 400 m to 1200 m in the Labrador and Irminger Basins, and a saltier, less well-oxygenated water mass observed from roughly 1500 m to 2300 m. These two water masses have been called Upper and Lower LSW, respectively (e.g., Kieke et al., 2007), and reflect variability in the severity of heat loss and depth of convection in the Labrador
Sea (Yashayaev et al., 2003; 2007) most likely associated with differences in atmospheric forcing during different phases of the North Atlantic Oscillation (NAO; Dickson et al., 1996; Lazier et al., 2002).

At the three stations in the Labrador Sea, Upper LSW has a $\delta^{30}Si_{DSi}$ of around +2 ‰ and a DSi concentration of $< 10$ µM (Fig 3). Lower LSW has slightly lower $\delta^{30}Si_{DSi}$ values (around +1.5 ‰) and slightly higher DSi concentrations (~10-15 µM). These
differences could be due to a slightly greater proportion of regenerated silica in these deeper layers, less frequently and intensively penetrated by deep convection, or to differences in the preformed properties of Upper and Lower LSW, a result of convection to greater depths during the formation of Lower LSW.

Below LSW, the central Labrador Sea (STN 69) exhibits an exemplary "stacking" of the water masses contributing to NADW.
Specifically, an increase in salinity at 3000 m points to the presence of NEADW, a modified version of the eastern Atlantic overflow water mass ISOW (van Aken and de Boer, 1995) that has crossed into the western Atlantic at the Charlie-Gibbs Fracture Zone (van Aken, 2000). At this point, NEADW has a $\delta^{30}Si_{DSi}$ of +1.5 ‰ and a DSi concentration of about 15 µM,



values essentially equal to that of Lower LSW since ISOW has entrained LSW during its journey from the Iceland-Scotland sill. At the very base of the water column, beginning at about 3500 m water depth, a decrease in salinity and increase in $O_2$ saturation point to the presence of DSOW, which flows from the Arctic Ocean into the North Atlantic as a bottom-hugging overflow off the eastern coast of Greenland, and represents the densest water contributing to NADW (Dickson and Brown,

1994). Our $\delta^{30}Si_{DSi}$ sample is situated within the transition between NEADW and DSOW, with slightly lower DSi than in NEADW, but no distinguishable difference in terms of its $\delta^{30}Si_{DSi}$ value of around +1.5 ‰.

Moving eastwards along the GEOVIDE transect brings us next to STN 60, on the southeastern coast of Greenland. This relatively shallow station has sampled IrSPMW at its upper two depths, which have high $\delta^{30}Si_{DSi}$ values (2.8 ‰ at 1000 m and

1.8 ‰ at 1400 m) and relatively low DSi concentrations (around 10 μM), reflective of the mixing of nutrient-depleted surface waters into this water mass. Subpolar Mode Waters can be seen as precursors to LSW, as they form pycnostads of progressively greater density from east to west in the subarctic gyre, preconditioning the upper water column for deep-reaching Labrador Sea convection by producing a relatively unstratified water column (Brambilla and Talley, 2008). Their more elevated $\delta^{30}Si_{DSi}$ values than that of LSW imply that the entrainment of DSi during deep convection associated with LSW formation plays an

important role in setting the final $\delta^{30}Si_{DSi}$ signature of this water mass. The deepest depth sampled at STN 60 (1800 m), on the other hand, is probably LSW, with its low PV, high $O_2$, and $\delta^{30}Si_{DSi}$ around +1.4 ‰, similar to the value observed in the central Labrador Sea.

Moving all the way across the Atlantic to the easternmost portion of the GEOVIDE transect allows us to focus in on two more

important interior water masses. One of the most striking features in the distributions of salinity and $O_2$ is the tongue of salty, $O_2$-poor water extending westward from the Iberian margin at about 1000 m water depth. This is predominantly MW that has entered the Atlantic through the Strait of Gibraltar (Iorga and Lozier, 1999). The one sample we have of predominantly MW is at 1000 m depth at STN 01, just off of the Iberian Peninsula, with a $\delta^{30}Si_{DSi}$ of +1.2 ‰ and a DSi concentration of about 10 μM. The $\delta^{30}Si_{DSi}$ value of +1.2 ‰ corresponds well to the value of +1.3 ‰ measured by Coffineau (2013) in samples from

closer to the point of origin of this water mass.

The other water mass sampled for $\delta^{30}Si_{DSi}$ in the eastern Atlantic Ocean is an $O_2$-poor, low-PV, Si-rich abyssal water mass that is present below about 3000 m. This is LDW (McCartney, 1992), which derives from northward-flowing AABW that has entered the eastern Atlantic Ocean via the Vema Fracture Zone (Mantyla and Reid, 1983; McCartney et al., 1991). The OMP

results of García-Ibáñez et al. (2017; this issue) shown in Fig. 5d nicely illustrate that LDW (which they denote as $NEADW_L$) is the dominant contributor to the DSi inventory of the deep eastern Atlantic. This is a direct result of LDW being rich in DSi (25-45 μM) when compared to the other water masses in the North Atlantic Ocean.



The influence of this Si-rich southern-sourced water mass is also clearly seen in the $\delta^{30}Si_{DSi}$ distribution (Fig 3b): $\delta^{30}Si_{DSi}$ values below 3000 m in the far-eastern Atlantic (STNs. 01, 13 and 21) range from +1.0 ‰ to +1.3 ‰, significantly lower than values at similar depths in the western Atlantic, which is dominated by northern-sourced water masses. The low $\delta^{30}Si_{DSi}$ values we observe for LDW compare very well with the value of +1.2 ‰ observed in AABW in the South Atlantic (de Souza et al.,

2012b), indicating that the Si-richness of this water mass makes its $\delta^{30}Si_{DSi}$ value insensitive to mixing or opal dissolution as it flows northwards in the abyssal Atlantic Ocean.

The remaining three stations in the mid-Atlantic (22−38°W; STNs. 26, 32 and 44) are influenced by varying combinations of the water masses of northern and southern origin that were discussed above. The easternmost of these three stations, STN 26

at the edge of the Porcupine Abyssal Plain, provides an exemplary illustration of the interaction of these water masses. At depths of 1400 and 2000 m, the water column at this station is dominated by Lower LSW, as reflected by the PV, $O_2$ and salinity distributions (Fig. 5a, b, c). As in the Labrador Sea itself, Lower LSW bears an elevated $\delta^{30}Si_{DSi}$ value, here about +1.7 ‰, and a relatively low DSi concentration of ~15 µM. This Si-poor water mass is underlain, at the very bottom of the profile (3500 m) by the Si-rich southern-sourced LDW (45 µM DSi) that bears a typically low $\delta^{30}Si_{DSi}$ value of +1.1 ‰.

The influence of dense LDW does not extend further west than the Porcupine Abyssal Plain, and thus at stations 44 and 32, Upper and Lower LSW give way to the denser ISOW (or its modified product, NEADW) with depth. As it flows over the Iceland-Scotland Ridge, ISOW mixes with more saline waters of the Atlantic thermocline to form NEADW that, being quite dense, comes to lie below LSW as it flows geostrophically along the western edge of the West European Basin (Fig. 1). The

differences in DSi concentration and $\delta^{30}Si_{DSi}$ between these northern-sourced water masses are small: ISOW-influenced waters at depths of 2500-3000 m bear values of +1.4 to +1.5 ‰ for $\delta^{30}Si_{DSi}$ and 15-25 µM for DSi, whilst LSW is only slightly more DSi-poor and correspondingly higher in $\delta^{30}Si_{DSi}$ (10-12 µM and +1.4 to +1.9 ‰, respectively). The resemblance in DSi concentration and $\delta^{30}Si_{DSi}$ between these two water masses is also due to the entrainment of LSW into ISOW. Interestingly, the very base of the water column at STN 44 is occupied by the dense and Si-poor DSOW, where it has a dissolved silicon

concentration of about 8 µM and a $\delta^{30}Si_{DSi}$ of +1.2 ‰. Although the DSi is typical for DSOW, the $\delta^{30}Si_{DSi}$ value of +1.2 ‰ is unexpectedly low for this water mass.

### 4.3 The influence of Labrador Sea Water on the North Atlantic distribution of $\delta^{30}Si_{DSi}$

The most important isotope fractionation signal in marine DSi is produced by diatom DSi uptake in the surface ocean (De La Rocha et al., 1997; Varela et al., 2004; Sutton et al., 2013), due to the dominant importance of these phytoplankton for the

marine Si cycle (Tréguer and De La Rocha, 2013; Hendry and Brzezinski, 2014). As a result, elevated values of $\delta^{30}Si_{DSi}$ can be produced only within the well-lit surface ocean, where photosynthesising organisms can grow and silicify. This surface-ocean signal is communicated more broadly by the process of water mass subduction, i.e., the transport of surface water parcels





into the ocean interior (Stommel, 1979). This is seen particularly clearly in our dataset, which spans a region in which exceptionally deep winter convection gives rise to mixed layers over 1 km deep in the Labrador Sea and Irminger Sea, injecting isotopically fractionated DSi into the ocean interior.

As can be seen from Fig. 5, there is a clear association of elevated $\delta^{30}Si_{DSi}$ values with the low-PV and high-$O_2$ signal of LSW, the water mass that is produced by deep winter convection. Indeed, the eastward spread of these elevated values coincides remarkably well with the extension of LSW mapped by McCartney and Talley (1982) based on PV (as shown in Fig. 1). The influence of LSW on the North Atlantic $\delta^{30}Si_{DSi}$ distribution is also nicely illustrated by the depth profiles in Fig. 3b, which show that, unlike the eastern Atlantic with low $\delta^{30}Si_{DSi}$ values at mid-depths, the central and western North Atlantic bears

elevated $\delta^{30}Si_{DSi}$ values close to those observed within the Labrador Sea itself. Such high values at mid-depths are unique to the North Atlantic Ocean amongst the major open-ocean basins, and result from the local formation of deep waters from Si-depleted surface waters of the subpolar North Atlantic.

Thus, one proximal physical control on the North Atlantic $\delta^{30}Si_{DSi}$ distribution is the vertical transport of DSi from the surface

ocean to mid-depths during LSW formation. Another physical control is shown by the close correlation between elevated $\delta^{30}Si_{DSi}$ and lower PV even within the eastern Atlantic Ocean, far from the region of deep convection (Fig. 5). This highlights the fact that the spreading of LSW as a result of the regional circulation transports its isotopic signal within the ocean interior, resulting in mid-depth $\delta^{30}Si_{DSi}$ values around +1.5 ‰ in regions where the physical signatures of LSW can be seen, documenting the importance of water mass structure on the marine $\delta^{30}Si_{DSi}$ distribution. Furthermore, this circulation pattern

results in the direct interaction of this northern-sourced water mass with the southern-sourced LDW, producing strong local $\delta^{30}Si_{DSi}$ gradients whose systematics correspond nicely to the basin-scale systematics (60°S to 60°N) documented by de Souza et al. (2012b; see Section 4.1 and Fig. 4).

Elevated values of $\delta^{30}Si_{DSi}$ are also associated with the dense overflows from the Nordic Seas. Whilst our single sample of

DSOW surprisingly bears a low $\delta^{30}Si_{DSi}$ value of +1.2 ‰, ISOW and its derivative NEADW bear similarly low DSi concentrations and similarly elevated $\delta^{30}Si_{DSi}$ values as LSW, reaching up to +1.5 ‰ in the abyssal Labrador Sea. They originate as dense bottom-hugging overflows of mid-depth Nordic Sea waters, influenced by the Arctic Ocean, that enter the North Atlantic across the submarine sills running between Greenland, Iceland and Scotland, and the elevated $\delta^{30}Si_{DSi}$ values of ISOW and NEADW reflect the isotopically-heavy nature of the deep Arctic (Varela et al. 2016). Both Brzezinski and Jones

(2015) and Varela et al. (2016) suggest that this feature results from the nature of the inflows to the Arctic Ocean, which receives isotopically fractionated DSi via the upper-ocean inflows from the Atlantic (and, to a lesser extent, the Pacific) due to the shallow sills that form its boundaries to these ocean basins. Observational and modelling studies indicate that these inflows are isotopically heavy primarily due to isotope fractionation during diatom DSi uptake in the Southern Ocean, although



more proximal fractionation within the Atlantic and Pacific Oceans most likely also plays some role (de Souza et al., 2012a; 2015).

Finally, interesting insights may be gained from a comparison of our Labrador Sea data with the only other published data

from this region (de Souza et al., 2012b). Fig. 6 compares data from the central Labrador Sea (STN 69) from the GEOVIDE study with literature data from within the Labrador Sea and slightly further south, in the vicinity of the Grand Banks (de Souza et al., 2012b), tracing LSW and NADW as they flow southwards. The three profiles agree within uncertainty at mid-depths and below, but diverge in the upper ocean at depths associated with Upper LSW. Since deep winter convection occurs up to depths of 1000 – 1500 m regularly within the Labrador Sea, this water mass is frequently ventilated locally, which may result

not only in variable physical properties (as shown in Fig. 5a,b,c) but also changes in its chemical characteristics, such as $\delta^{30}Si_{DSi}$. However, care should be taken not to over-interpret such differences of ~0.3 ‰, given the potential for $\delta^{30}Si_{DSi}$ offsets between laboratories of ±0.2 ‰, as discussed in Section 4.1 (Reynolds et al., 2007; Grasse et al. 2017).

## 5 Conclusion

Water mass subduction and circulation appears to be the dominant process influencing the distribution of DSi in the North

Atlantic Ocean and Labrador Sea. Our dataset of $\delta^{30}Si_{DSi}$ along the GEOVIDE transect documents the extent to which the distribution of $\delta^{30}Si_{DSi}$ in the North Atlantic Ocean and Labrador Sea is influenced by the hydrography of this region. At depths below 1000 m, the distribution of $\delta^{30}Si_{DSi}$ is clearly linked to water mass structure, with the two dominant influences coming from northern-sourced waters (LSW and ISOW) and southern-sourced waters (LDW). The Si-poor northern-sourced waters impart the intermediate and mid-depth North Atlantic Ocean with elevated $\delta^{30}Si_{DSi}$ values over +1.4 ‰ and up to +1.9 ‰,

whilst the Si-rich abyssal LDW results in low $\delta^{30}Si_{DSi}$ values of +1.1 ‰ to +1.3 ‰ in the deepest eastern Atlantic Ocean. By combining our isotope data with hydrographic information and results from an optimum multiparameter analysis, we show that the $\delta^{30}Si_{DSi}$ distribution bears clear evidence of the influence of LSW flowing across the Atlantic Ocean into the eastern basins, in a manner consistent with McCartney and Talley's (1982) canonical map of the extent of this water mass. As a result, the eastern Atlantic exhibits the direct "stacking" of young, Si-poor LSW above old, Si-rich LDW, producing a range in deep

ocean $\delta^{30}Si_{DSi}$ values within this one ocean basin that is comparable to that observed over the entire latitudinal range of the Atlantic Ocean and, indeed, in the global deep ocean.

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





**Table 1.** Sampling locations from the GEOVIDE voyage in the North Atlantic Ocean and Labrador Sea (Stations 1-77) and GEOVIDE/Canada GEOTRACES intercalibration stations (Stations K1 and LS2).

| Station | Latitude (°N) | Longitude (°E) |
|---------|---------------|----------------|
| 1 | 40.333 | -10.036 |
| 13 | 41.383 | -13.888 |
| 21 | 46.544 | -19.672 |
| 26 | 50.278 | -22.602 |
| 32 | 55.506 | -26.71 |
| 44 | 59.623 | -38.954 |
| 60 | 59.799 | -42.003 |
| 64 | 59.068 | -46.083 |
| 69 | 55.842 | -48.093 |
| 77 | 53.000 | -51.100 |
| K1 | 56.197 | -53.425 |
| LS2 | 60.593 | -56.589 |

5 **Table 2. Mass spectrometer operating conditions**

| | |
|---|---|
| *Resolution* | Medium |
| *Forward Power* | 1200 W |
| *Accelerating Voltage* | 10 kV |
| *Plasma Mode* | Dry Plasma |
| *Cool Gas Flow Rate* | 15.5 L min$^{-1}$ |
| *Auxiliary Gas Flow Rate* | 0.8 L min$^{-1}$ |
| *Sample Gas Flow Rate* | ~1 L min$^{-1}$ |
| *Sampler Cone* | Standard Ni cone |
| *Skimmer Cone* | Standard Ni cone |
| *Desolvator* | Apex (ESI) |
| *Nebulizer* | PFA microcentric nebuliser 60 μL min$^{-1}$ |
| *Running Concentrations* | Si = 1-2 ppm, Mg = 1-2 ppm |
| *Sensitivity* | 7-10 V ppm$^{-1}$ |
| *Blank Level* | < 1 % signal |
| *$^{30}Si$ Interference* | < 30 mV (usually 10-15 mV) |





**Table 3.** Results of GEOVIDE/Canada GEOTRACES intercalibration exercise for $\delta^{30}Si_{DSi}$ (‰). Following the GEOTRACES intercalibration protocol for a cruise without a cross-over station (i.e., duplicate samples at 3 depths and 2 stations; see Table 1 for location details), the mean $\delta^{30}Si$ (± 2 standard deviations; 2SD) was analysed by two separate laboratories (ETH Zurich and Ifremer Plouzané). The difference in $\delta^{30}Si$ between the laboratories (values determined for ETH Zurich minus values determined for Ifremer/UBO) is also presented.

|  |  |  | ETH Zurich | | Ifremer Plouzané | | Difference |
|---|---|---|---|---|---|---|---|
|  | Depth | [Si] | $\delta^{30}Si$ (‰) | | $\delta^{30}Si$ (‰) | | $\delta^{30}Si$ (‰) |
| Station | (m) | μM | mean | 2SD | mean | 2SD | |
| K1 | 300 | 7.79 | **1.95** | 0.16 | **1.89** | 0.17 | **0.06** |
| K1 | 500 | 8.06 | **1.79** | 0.14 | **1.90** | 0.14 | **-0.11** |
| K1 | 1000 | 8.34 | **1.74** | 0.18 | **1.76** | 0.16 | **-0.02** |
| LS2 | 100 | 6.77 | **1.97** | 0.24 | **1.92** | 0.14 | **0.05** |
| LS2 | 2000 | 9.92 | **1.68** | 0.12 | **1.68** | 0.16 | **0.00** |
| LS2 | 3000 | 11.92 | **1.72** | 0.12 | **1.66** | 0.14 | **0.06** |

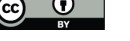

**Figures**

**Figure 1.** Map showing the sampling locations from the GEOVIDE voyage, overlain on a schematic of the intermediate and deep ocean circulation. Pink shading indicates the spreading of Labrador Sea Water (LSW) as documented by the extent of a
water column minimum potential vorticity of $8\times10^{-12}$ m$^{-1}$ s$^{-1}$ (McCartney, 1992). Dark blue arrows represent bottom-hugging Nordic Sea Overflows (ISOW and DSOW), pink arrows represent LSW, and orange arrows represent Lower Deep Water (McCartney, 1992; Dickson and Brown, 1994; Lambelet et al., 2016; see also García-Ibáñez et al., 2017; this issue).

**Figure 2.** Mass-dependent fractionation line of $\delta^{29}$Si vs. $\delta^{30}$Si (‰ vs. NBS28) for samples (n = 44) collected from 10 depth
profiles in North Atlantic Ocean and Labrador Sea. MDF line represented by $\delta^{29}$Si=0.52*$\delta^{30}$Si, R$^2$=0.99, 2 SD=0.16 ‰ $\delta^{30}$Si.

**Figure 3.** Depth profiles of (a) Si concentration and (b) $\delta^{30}$Si$_{DSi}$ values for the North Atlantic Ocean and Labrador Sea. Nutrient data collected during the GEOVIDE cruise from a separate cast are indicated as dashed lines (see Supplementary Table S1).

**Figure 4.** $\delta^{30}$Si$_{DSi}$ versus the inverse of DSi concentration for this study (black circles), de Souza et al. (2012b; grey circles) and Brzezinksi and Jones (2015; open circles) for waters > 1000 m depth. Equations reported in the figure refer to linear regressions produced for each dataset.

**Figure 5**. Depth section across the GEOVIDE transect with hydrographic parameters (a) salinity, (b) oxygen (O$_2$) saturation
and (c) potential vorticity (PV), together with (d) $\delta^{30}$Si$_{DSi}$ data overlain by pie charts of the fraction of DSi in each sample contributed by various water masses, as calculated by OMP analysis (García-Ibáñez et al., 2017; this issue). Spreading of Labrador Sea Water, reflected in hydrography by low-salinity, low-PV, high-O$_2$ signals (panels a,b,c) and in the OMP results by a dominant Si contribution from this water mass (panel d), produces a mid-depth extension of elevated $\delta^{30}$Si values into the eastern Atlantic (panel d). For water mass abbreviations see main text (Section 2.4).
**Figure 6.** Data from the Labrador Sea (STN 69; blue) during GEOVIDE cruise (2014) compared with data from samples collected in 2010 from nearby stations in the Labrador Sea (green) and at the Grand Banks (orange) (STNs. 8 and 11 in de Souza et al. (2012b).






**Acknowledgements**

The authors thank the UMS flotte, GENAVIR, DT INSU in the realization of the GEOVIDE mission and the captain and crew of the R/V *Pourquoi Pas?*. Special thanks to M. Le Goff, E. Grossteffan, K. Giesbrecht, and L. Foliot for helping with the sampling of water for this project and Emmanuel Ponzevera (Unité Géosciences Marines; Ifremer) for providing assistance

5  with the mass spectrometry. This work was supported by the "Laboratoire d'Excellence" LabexMER (ANR-10-LABX-19) and co-funded by a grant from the French government under the program "Investissements d'Avenir", and by a grant from the Regional Council of Brittany (SAD programme). The GEOVIDE project was funded by CNRS-INSU (programme LEFE), ANR "Blanc" GEOVIDE (ANR-13-BS06-0014) and "RPDOC" BITMAP (ANR-12-PDOC-0025), the LabexMER and Ifremer. Gregory de Souza was supported by a Marie Skłodowska-Curie Research Fellowship under EU Horizon 2020 (SOSiC;

10  #708407). Maribel I. Garcia-Ibáñez was supported by the Spanish Ministry of Economy and Competitiveness through the BOCATS (CTM2013-41048-P) project co-funded by the Fondo Europeo de Desarrollo Regional 2014–2020 (FEDER).





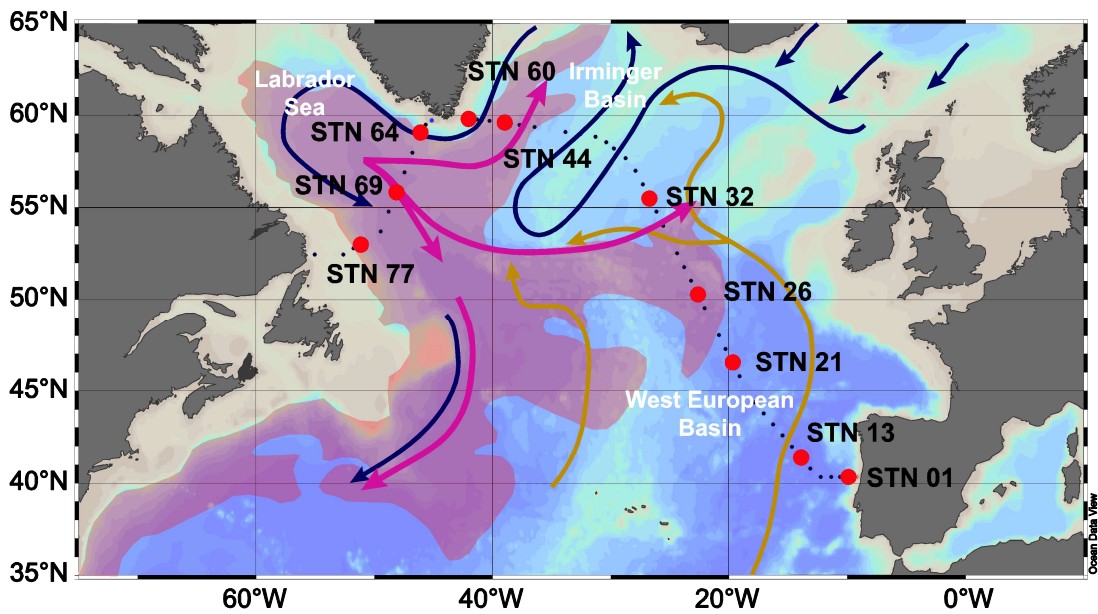





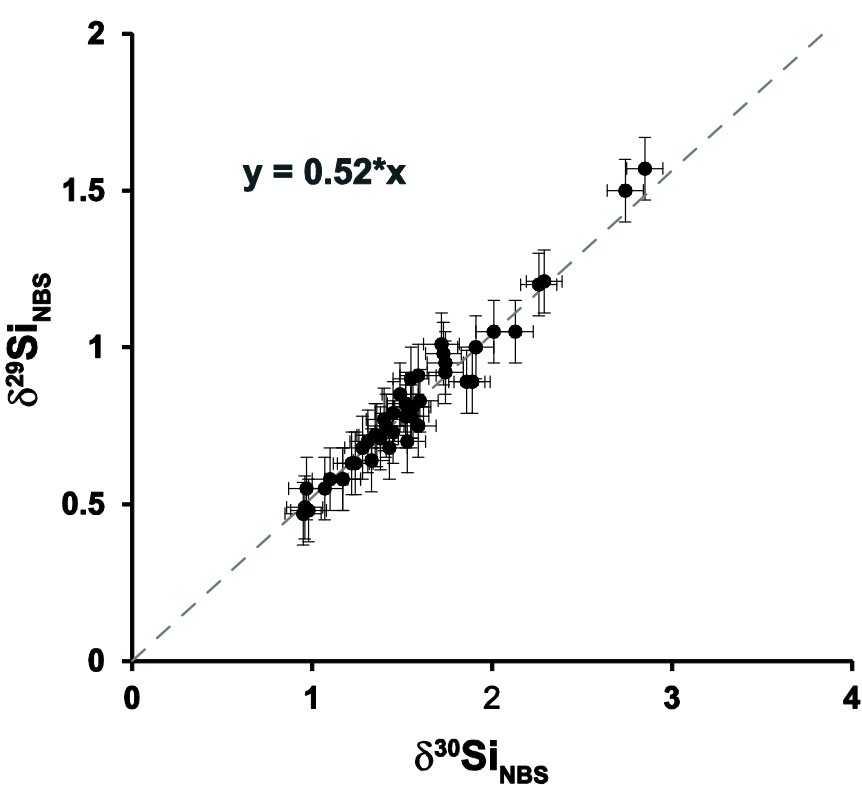



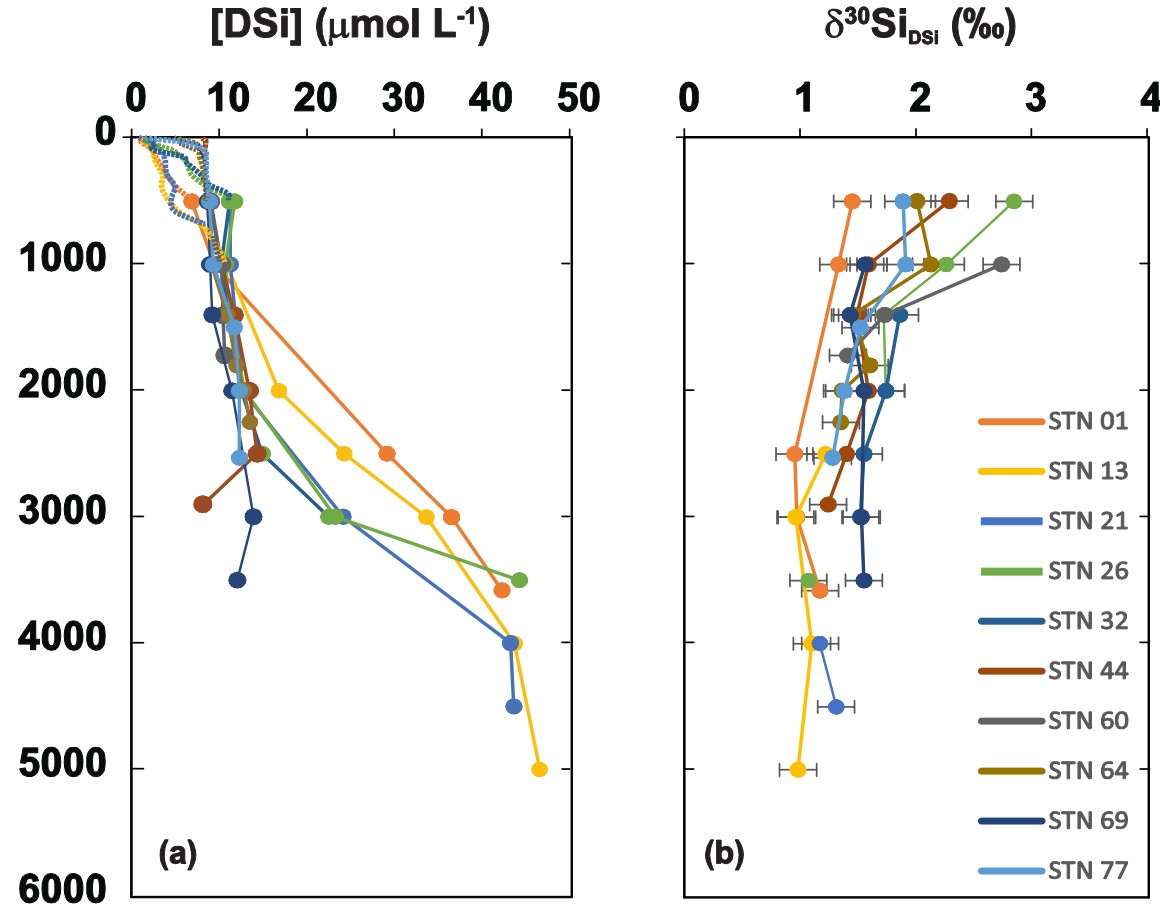





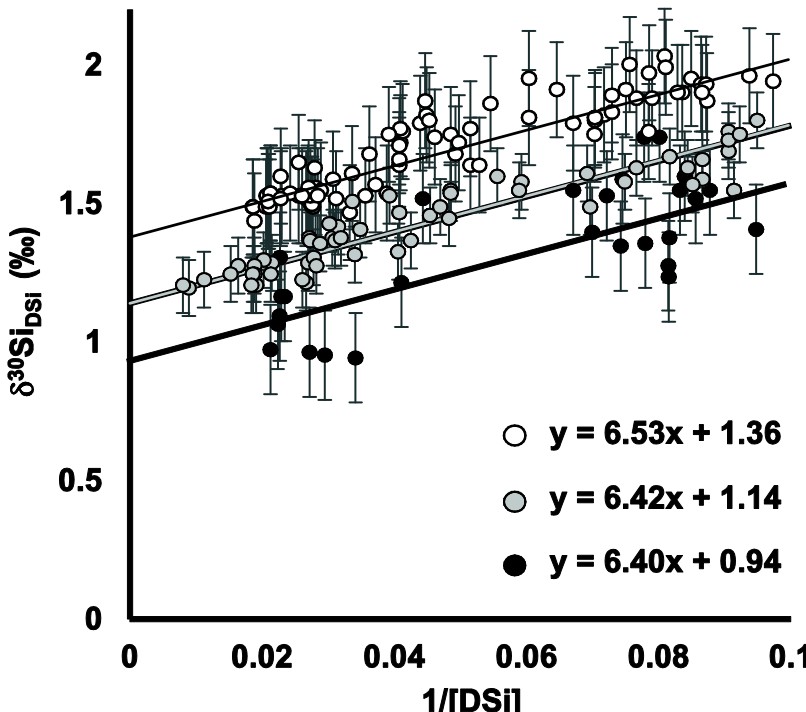





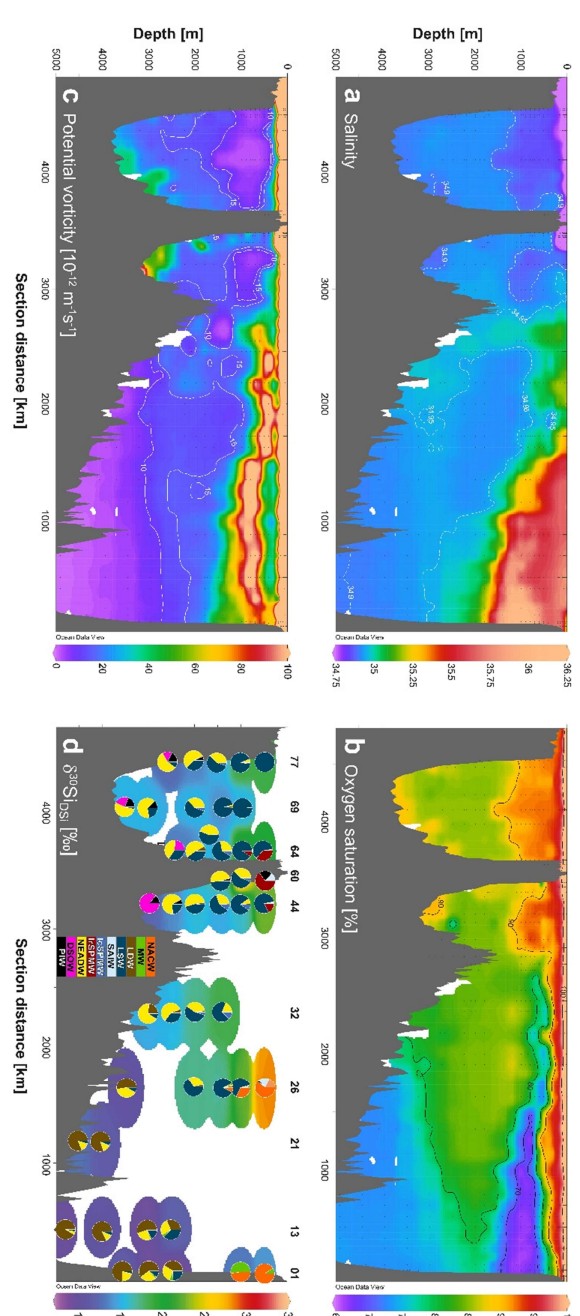




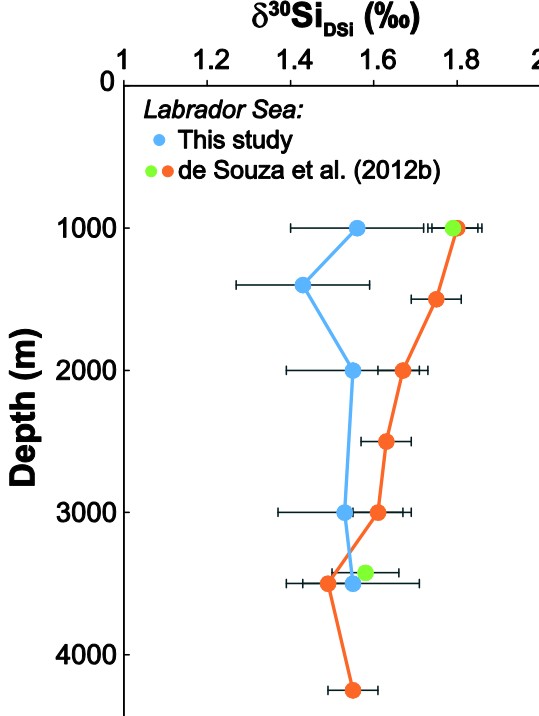