# Peer review of "The silicon stable isotope distribution along the GEOVIDE section of the North Atlantic Ocean"

_Biogeosciences, 2018_

## Referee Comment (RC1) · P. Grasse (Referee) · 8 May 2018

Review

The silicon isotope distribution along the GEOVIDE section of the North Atlantic Ocean

Jill N. Sutton et al.

The manuscript by Sutton et al. shows the dissolved silicon isotope ($\delta30dSi$) distribution along the GEOVIDE section in the North Atlantic Ocean. The authors present 10 Stations along the transect. The data is of high quality and intercalibrated according to GEOTRACES protocol. Samples in the upper 500 m could not be measured due to low DSi. The manuscript discusses in detail the influence of water masses on the $\delta30dSi$. However, some parts of the water mass discussion need to be improved.

due to low DSi. The manuscript discusses in detail the influence of water masses on the $\delta30dSi$. However, some parts of the water mass discussion need to be improved.

The discussion would benefit from a plot with water mass end members to identify all sources and possibly other processes influencing $\delta$30dSi in intermediate and deep waters. The manuscript is overall very good and suitable for Biogeociences, but needs revision (medium) to improve the discussion section. Please find my comments below.

Abstract:

P1 L12: I think the information that low DSI samples could not be measured does not necessarily need to be in the Abstract. I think it would be more important that this information is mentioned in the methods or results part.

Methods:

P3 L20: Mesh size of the AG 1 X8 resin? P4 L20: Please check the equation. Shouldn't it be 26/24 Mg or 25/24Mg? P5 L15: It seems the manuscript Garcia-Inánez et al. is already accepted.

Results:

P6 L5: Please include one sentence about the DSI concentrations in the upper 500 m and that the samples could not be analyzed.

P6 L8: I would add some information to describe the DSi and $\delta$30dSi in more detail. E.g. St. 1 and St. 13 already increase at 1000 m depth.

P6 L18: Please give the exact values of the lowest $\delta$30dSi (0.95 ‰ to 0.98 ‰. I think these very low $\delta$30dSi values are actually quite interesting and need some more attention. See my comments below.

Discussion: P7 L13: The constant offset between the data set from Brzezinski and Jones and de Souza et al., must be indeed a measurement artifact. However, the offset between your data set and de Souza is partly bigger than 0.2 ‰ and samples of the GEOVIDE section do not seem to plot on a straight line between DSi and $\delta$30dSi. It looks that the data clusters as some samples show a wide range in $\delta$30dSi (1 ‰ to 1.7

‰ at nearly similar concentrations (approximately 40 $\mu$mol) and at low DSi (appr. 12 $\mu$mol) where $\delta$30dSi ranges from 1.25 ‰ to 1.7 ‰That could indicate that other sources or processes influence the waters of your study compared to the open ocean stations in de Souza at al., and Brzezinski and Jones. I think it would be helpful to modify figure 4. First of all, you should make your data more visible (e.g., bring your data to the front, use a light color for the already published data). You could try to group your data. e.g., only use open ocean stations vs. stations close to landmasses. Colorcode the stations or samples that are characterized by specific water masses. It would also be helpful to add water mass end members, e.g., AABW, which brings a light source from the south 1.2 ‰ (0.01 DSi; Souza et al. 2012). That could show additional processes that influence your deep-water masses e.g. at St. 1 and St. 13. Generally, I think it is interesting, that you see such light $\delta$30dSi values and it should be discussed in more detail. According to your intercalibration with de Souza et al. (Fig. 6) and your results from the intercalibration study Grasse et al. (2017) your $\delta$30dSi data agrees very well within error (0.1 ‰ 2sd). Therefore, a water sample of 1 ‰ together with slightly higher DSi compared to de Souza et al., might indicate that further remineralization influences the $\delta$30dSi composition. Such low (or even lower, 0.6 ‰ values are typically associated with much higher DSi of 130 to 150 micromol in the Pacific and (Reynolds et al.2006, de Souza et al., 2012, Grasse et al., 2013) at DSi concentrations (even though I know that some people doubt some of the $\delta$30dSi deep water values in the North Pacific). However, Grasse et al. 2016 observed $\delta$30dSi values of 1.1 ‰ in bottom water of the Peruvian shelf (âĹij40 micromol), which were influenced by pore waters from the sediment and remineralization at the sediment-seawater interface (Ehlert et al., 2016). Not necessary an effect you observe, but if not dissolution at the seawater-sediment interface or in the water column influences your $\delta$30dSi, you could also have admixture with a distinct water mass that brings in a very light $\delta$30dSi signature (e.g., a water masses from Iceland? I am not so familiar with the water mass circulation in the Atlantic, but it seems that the NEADW can pick up its signature here?). Additionally, the circulation is quite sluggish, or? Therefore, you can have a trapping effect? I do not

want, that you go too much into detail into the Pacific seawater $\delta$30dSi distribution and I also see that some of the values are identical within error, but I would like to have a better explanation why not all of your data does fall on the line for DSi versus $\delta$30dSi.

P7 L8: Please give the values (low, high) for the study by de Souza et al.

P7 L25: please mention here (or at least above) the absolute $\delta$30dSi values from the study of de Souza et al. for comparison with your values. The range can be similar, but that does not necessarily mean, that the $\delta$30dSi are identical.

P10 L8: Please also explain, why the uppermost sample at station 26 has such high $\delta$30dSi.

P11 L5: Please mention the stations you are talking about. High $\delta$30dSi? Value? What values?

P11 L10: What are the $\delta$30dSi values in the Labrador Sea? Please make clear that it is subducted surface water.

P11 L24: Please give me the station number and depth that makes it much easier to follow and understand your discussion.

P11 L25 Doesn't NEADW has high DSi? Here I am getting confused, isn't the NEADW influencing the eastern deep waters? At least according to Fig 4. in Garcia-Ibanez et al.? Please check the Garcia-Ibanez paper for water masses; it seems that there are some discrepancies, most likely as a result of the review process of the manuscript.

Figures: Fig.2 I do not think that the Figures has to be in the Paper. In my opinion, it is enough to mention in the text, that all samples fall on the mass-dependent fractionation line.

Fig4: Can you please adjust the y-scale from 0.5 ‰ to 2 ‰ Please add the studies indicated by different color directly to the legend. Would be good to modify the figure (see comments above)

Fig. 5: It is quite tricky to distinguish the colors of different water mass types. You could only name the dominant water mass in the figure. Similar to Garcia-Ibanez et al. (Figure 4). Can you replace section distance with longitude?

References: de Souza, G. F., Reynolds, B. C., Johnson, G. C., Bullister, J. L., & Bourdon, B. (2012). Silicon stable isotope distribution traces Southern Ocean export of Si to the eastern South Pacific thermocline. Biogeosciences, 9(1), 4199–4213.

Grasse, P., Ehlert, C., & Frank, M. (2013). The influence of water mass mixing on the dissolved Si isotope composition in the Eastern Equatorial Pacific. Earth and Planetary Science Letters, 380, 60–71.

Grasse, P., Ryabenko, E., Ehlert, C., Altabet, M. A., & Frank, M. (2016). Silicon and nitrogen cycling in the upwelling area off Peru: A dual isotope approach. Limnology and Oceanography, 61(5), 1661–1676.

Ehlert, C., Doering, K., Wallmann, K., Scholz, F., Sommer, S., Grasse, P., et al. (2016). Stable silicon isotope signatures of marine pore waters – Biogenic opal dissolution versus authigenic clay mineral formation. Geochimica Et Cosmochimica Acta, 191, 102–117.

Reynolds, B., Frank, M., & Halliday, A. (2006). Silicon isotope fractionation during nutrient utilization in the North Pacific. Earth and Planetary Science Letters, 244(1-2), 431–443.

---

## Short Comment (SC1) · 27 May 2018

Review of bg 2018 165

This paper presents dissolved Si isotopic data of intermediate and deep water masses in the North Atlantic. This is a key area for the Meridional Overturning Circulation. The GEOVIDE section has sampled 10 stations analysed in this study that are particularly relevant. The results presented here is the third study on the North Atlantic. It increases the offset found by Brzezinski & Jones (2015) compared to de Souza et al. 2012 in Si isotopic data in the region (from +/- 0.1 pmil to +/- 0.2 pmil), which was unexplained by the recent intercalibration published (Grasse et al. 2017). To consolidate their data, the authors have made an inter-comparison of 6 GEOVIDE samples (as required by

GEOTRACES protocol) with ETH Zurich and the results compare very well. Even though the cause of the offset among the three North Atlantic studies remains still unsolved, this makes their data trustable.

The paper is very well written and concise. The results are discussed in terms of mixing of the high number of water masses present along the transect since the authors generally did not find a significant imprint of dissolution of biogenic silica at depth.

Therefore I recommend publication of this work and I've listed below my few minor to moderate comments on the paper.

Damien Cardinal

- P4 L2 vs. Table 1: in the text neb flow rate is 100 uL/min while it is 60 uL/min in Table 1. Homogenise.

- Fig. 4 and in the text associated. 1) In this figure, the authors compare their GEOVIDE data with the two previous studies in the North Atlantic of Brzezinski & Jones (2015) and de Souza et al. (2012). Since Brzezinski & Jones chose to correct the offset between their data and the ones of de Souza et al. by +- 0.11 pmil, I suggest the authors here clearly mention that they always use the non-corrected data (which I believe is the right way to proceed) to avoid confusion with corrected data discussed in Brzezinski & Jones. 2) Important. Provide error bars of the three slopes and intercepts. Variability of GEOVIDE dataset seems higher. This should be checked and discussed. It is particularly needed given the offset found between the three data set that remains unsolved.

- Fig. 5d is a key figure and is much too small when printed. Moreover the DSi concentration is missing. I suggest to restrict Fig. 5 to the current panels a, b, c and to add a fig. 6 with current panel 5d + a panel with DSi concentration. Alternatively, Fig. 5 could cover a full A4 page and not just less than half of it.

- Could the authors provide a table with d30Si and DSi end-members of water masses

as calculated from their isotopic data and the contribution based on OMP from Garcia-Ibanez et al. (2017)? This would be very useful.

- Supplementary Table S1: provide in the Table caption the definition of Si* = DSi – NO3

---

## Short Comment (SC2) · 28 May 2018

Thank you for the detailed comments, they are very useful and will help with the revision of our manuscript.

I want to clarify a few questions/comments made by the reviewer, since I am not sure if I understood them correctly.

Firstly, under the sub-header Discussion, a comment was made about providing further discussion on why we see light d30Si values. I agree that we can provide more discussion here, and I like your suggestions. To quickly answer, due to the formation history of NEADW, we can pick up the light 30dSi signature from water masses from Iceland, since NEADW derives from ISOW. We briefly touched on this in the manuscript (p5

L31-p6 L2): "NEADW is formed as a result of entrainment events that occur along the journey of ISOW through the Iceland Basin (van Aken, 2000). NEADW recirculates in the West European Basin and mixes with the surrounding waters, including the Antarctic Bottom Water (AABW) (van Aken and Becker, 1996), resulting in the formation of LDW". We can expand in further detail again in the discussion. Also, the circulation within the West European Basin is slow, and NEADW is a relatively old water mass (see attached paper Fleischmann et al., 2001).

Secondly, for the comment P11 L25 regarding DSi in NEADW. I can understand the confusion but we had included some text in the manuscript to try to clarify this point (see P9 L29 : "The OMP results of García-Ibáñez et al. (2017; this issue) shown in Fig. 5d nicely illustrate that LDW (which they denote as NEADWL) is the dominant contributor to the DSi inventory of the deep eastern Atlantic"). As answered in the previous comment, NEADW is derived from ISOW so it is consistent to have similar d30Si in both ISOW and NEADW. And, you are correct, NEADW is the dominant water mass for the deep Eastern area of OVIDE. However, as mentioned above (and in the text), NEADW recirculates in the West European Basin and mixes with the surrounding waters, including the Antarctic Bottom Water (AABW) (van Aken and Becker, 1996), resulting in the formation of LDW. The concentration of DSi in the NEADW is not very high (i.e. 15 uM) but the LDW does have a higher concentration (25-45 uM) since it mixes with AABW. So, I think the main point of confusion is this: Garcia-Ibanez et al. refer to the LDW as the NEADWL. We decided to avoid using the NEADWL for two reasons: (1) to clearly separate NEADW and NEADWL (in our text we describe it as LDW), and (2) because most people refer to the second water mass as LDW.

Please also note the supplement to this comment:
https://www.biogeosciences-discuss.net/bg-2018-165/bg-2018-165-SC2-supplement.pdf

**Supplement:**

[supplement omitted: unrelated document]

---

## Short Comment (SC3) · 28 May 2018

Thank you for the review and your comments. They all seem appropriate and we are happy to make the revisions.

---

## Author Comment (AC1) · 14 Jun 2018

Dear Damien Cardinal,

As requested by Biogeosciences, I need to individually respond to all referee comments (RCs) by posting final author comments on behalf of all co-authors. Therefore, I will reiterate my response to your review below.

Thank you for very much for your review and your constructive comments. They all seem appropriate and we are happy to make the revisions.

Sincerely, Jill Sutton

---

## Author Comment (AC2) · 14 Jun 2018

Dear Patricia Grasse,

As requested by Biogeosciences, I need to individually respond to all referee comments (RCs) by posting final author comments on behalf of all co-authors.

Again, thank you for very much for your review and your constructive comments. We are happy to make most of the changes that you suggest. In particular, we will try to incorporate the changes that you suggest for Figure 4 and provide a more detailed discussion, as previously mentioned in our comment during the open discussion part of this review process.

Sincerely, Jill Sutton

---

## Author Comment (AC3) · 27 Jun 2018

We would again like to thank Dr. Patricia Grasse for their thorough review of our manuscript and their helpful comments. We believe that we can address all of the major comments indicated by Dr. Grasse as indicated in the discussion below. Note that the italicized text represents the comments made by Dr. Grasse and non-italicized/bold text is our response.

*P1 L12: I think the information that low DSI samples could not be measured does not necessarily need to be in the Abstract.   I think it would be more important that this information is mentioned in the methods or results part.* **We will make the changes recommended by the reviewer.**

*P3 L20: Mesh size of the AG 1 X8 resin?*  **We will add the text "(100-200 mesh size)"**

*P4 L20: Please check the equation. Shouldn't it be 26/24 Mg or 25/24Mg?*  **No, it does not matter if we use 25/26, 26/24, or 25/24 since this is used as a technique to normalize the data. The ratios should be constant and consistent and any fractionation observed is due to the effects of mass spectrometry. This allows us then to normalize the silicon isotope data.**

*P5 L15:  It seems the manuscript Garcia-Inánez et al.  is already accepted.* **This was not the case at the time of submission and will be changed.**

*P6 L5: Please include one sentence about the DSI concentrations in the upper 500 m and that the samples could not be analyzed.* **We will make the changes recommended by the reviewer.**

*P6 L8:  I would add some information to describe the DSi and δ30dSi in more detail. E.g. St. 1 and St. 13 already increase at 1000 m depth.* **We will make the changes recommended by the reviewer.**

*P6 L18:  Please give the exact values of the lowest δ30dSi (0.95 ‰ to 0.98 ‰. I think these very low δ30dSi values are actually quite interesting and need some more attention. See my comments below.* **We will make the changes recommended by the reviewer.**

*P7 L13:  The constant offset between the data set from Brzezinski and Jones and de Souza et al., must be indeed a measurement artifact.   However, the offset between your data set and de Souza is partly bigger than 0.2 ‰ and samples of the GEOVIDE section do not seem to plot on a straight line between DSi and δ30dSi. It looks that the data clusters as some samples show a wide range in δ 30dSi (1 ‰ to 1.7 C2 ‰ at nearly similar concentrations (approximately 40 μmol) and at low DSi (appr.  12 μmol) where δ30dSi ranges from 1.25 ‰ to 1.7 ‰. That could indicate that other sources or processes influence the waters of your study compared to the open ocean stations in de Souza at al., and Brzezinski and Jones. I think it would be helpful to modify figure 4.  First of all, you should make your data more visible (e.g., bring your data to the front, use a light color for the already published data). You could try to group your data. e.g., only use open ocean stations vs.  stations close to landmasses.  Colorcode the stations or*

*samples that are characterized by specific water masses. It would also be helpful to add water mass end members, e.g., AABW, which brings a light source from the south 1.2 ‰ (0.01 DSi; Souza et al. 2012). That could show additional processes that influence your deep-water masses e.g. at St. 1 and St. 13. Generally, I think it is interesting, that you see such light δ30dSi values and it should be discussed in more detail. According to your intercalibration with de Souza et al. (Fig. 6) and your results from the intercalibration study Grasse et al. (2017) your δ 30dSi data agrees very well within error (0.1 ‰ 2sd). Therefore, a water sample of 1 ‰ together with slightly higher DSi compared to de Souza et al., might indicate that further remineralization influences the δ30dSi composition. Such low (or even lower, 0.6 ‰ values are typicallyassociated with much higher DSi of 130 to 150 micromol in the Pacific and (Reynolds et al.2006, de Souza et al., 2012, Grasse et al., 2013) at DSi concentrations (even though I know that some people doubt some of the δ 30dSi deep water values in the North Pacific). However, Grasse et al. 2016 observed δ30dSi values of 1.1 ‰ in bottom water of the Peruvian shelf (â Li j40 micromol), which were influenced by pore waters from the sediment and remineralization at the sediment-seawater interface (Ehlert et al., 2016). Not necessary an effect you observe, but if not dissolution at the seawater-sediment interface or in the water column influences your δ 30dSi, you could also have admixture with a distinct water mass that brings in a very light δ 30dSi signature (e.g., a water masses from Iceland? I am not so familiar with the water mass circulation in the Atlantic, but it seems that the NEADW can pick up its signature here?). Additionally ,the circulation is quite sluggish, or? Therefore, you can have a trapping effect? I do not C3 want, that you go too much into detail into the Pacific seawater δ 30dSi distribution and I also see that some of the values are identical within error, but I would like to have a better explanation why not all of your data does fall on the line for DSi versus δ 30dSi*.

**Thank you for bringing up these important points, we will make the necessary changes to make this information clearer. I agree that we can provide more discussion here, and I like your suggestions. Due to the formation history of NEADW, we can pick up the light 30dSi signature from water masses from Iceland, since NEADW derives from ISOW. We briefly touched on this in the manuscript (p5 L31-p6 L2): "NEADW is formed as a result of entrainment events that occur along the journey of ISOW through the Iceland Basin (van Aken, 2000). NEADW recirculates in the West European Basin and mixes with the surrounding waters, including the Antarctic Bottom Water (AABW) (van Aken and Becker, 1996), resulting in the formation of LDW". We can expand in further detail again in the discussion. Also, the circulation within the West European Basin is slow, and NEADW is a relatively old water mass (see attached paper Fleischmann et al., 2001).**

**In addition, after I received the reviews, I noticed that one data point was missing from the supplementary table (although had been included in Fig.4) and that several data points were missing from Fig. 4 for the depth range between 1000-1500m. These data were available in the supplementary table, but I will now include them in the new Fig. 4.**

*P7 L8: Please give the values (low, high) for the study by de Souza et al.* **We will make the changes recommended by the reviewer.**

*P7 L25:  please mention here (or at least above) the absolute δ30dSi values from the study of de Souza et al. for comparison with your values. The range can be similar, but that does not necessarily mean, that the δ30dSi are identical.* **We will make the changes recommended by the reviewer.**

*P10 L8:  Please also explain, why the uppermost sample at station 26 has such high δ30dSi.* **We will make the changes recommended by the reviewer.**

*P11 L5: Please mention the stations you are talking about. High δ30dSi? Value? What values? P11 L10: What are the δ 30dSi values in the Labrador Sea?  Please make clear that it is subducted surface water.* **We will make the changes recommended by the reviewer.**

*P11 L24:  Please give me the station number and depth that makes it much easier to follow and understand your discussion.* **We will make the changes recommended by the reviewer.**

*P11 L25 Doesn't NEADW has high DSi? Here I am getting confused, isn't the NEADW influencing the eastern deep waters?  At least according to Fig 4.  in Garcia-Ibanez et al.?  Please check the Garcia-Ibanez paper for water masses; it seems that there are some discrepancies, most likely as a result of the review process of the manuscript.* **As mentioned in the response to reviewers (and in the text), NEADW recirculates in the West European Basin and mixes with the surrounding waters, including the Antarctic Bottom Water (AABW) (van Aken and Becker, 1996), resulting in the formation of LDW. The concentration of DSi in the NEADW is.  NEADW is the dominant water mass for the deep Eastern area of OVIDE. As answered in the previous comment, NEADW is derived from ISOW so it is consistent to have similar d30Si in both ISOW and NEADW.**

*Fig.2 I do not think that the Figures has to be in the Paper. In my opinion, it is enough to mention in the text, that all samples fall on the mass-dependent fractionation line.* **We will make the changes recommended by the reviewer and remove Fig. 2.**

*Fig4:  Can you please adjust the y-scale from 0.5 ‰ to 2 ‰  Please add the studies indicated by different color directly to the legend.  Would be good to modify the figure (see comments above) C4* **We will make the changes recommended by the reviewer.**

*Fig.  5:  It is quite tricky to distinguish the colors of different water mass types.  You could only name the dominant water mass in the figure. Similar to Garcia-Ibanez et al. (Figure 4). Can you replace section distance with longitude?* **We will make the changes recommended by the reviewer.**

---

## Author Comment (AC4) · 27 Jun 2018

We would again like to thank Professor Damien Cardinal for their thorough review of our manuscript and their helpful comments. We believe that we can address all of the comments indicated by D. Cardinal as indicated in the discussion below. Note that the italicized text represents the comments made by D. Cardinal and non-italicized/bold text is our response.

*P4 L2 vs. Table 1: in the text neb flow rate is 100 uL/min while it is 60 uL/min in Table 1. Homogenise.* **We will change the flow rate to 100 uL/min in Table 1.**

*- Fig. 4 and in the text associated. 1) In this figure, the authors compare their GEOVIDE data with the two previous studies in the North Atlantic of Brzezinski & Jones (2015) and de Souza et al. (2012). Since Brzezinski & Jones chose to correct the offset between their data and the ones of de Souza et al. by +- 0.11 pmil, I suggest the authors here clearly mention that they always use the non-corrected data (which I believe is the right way to proceed) to avoid confusion with corrected data discussed in Brzezinski & Jones. 2) Important. Provide error bars of the three slopes and intercepts. Variability of GEOVIDE dataset seems higher. This should be checked and discussed. It is particularly needed given the offset found between the three data set that remains unsolved.*

**Thank you for bringing up these important points. We will make the changes that you suggest. Indeed, the variability of the GEOVIDE data set is higher (stdev of the slope of 1.1 vs. 0.3 for the two other slopes). In addition, I noticed that one data point was missing from the supplementary table (although had been included in Fig.4) and that several data points were missing from Fig. 4 for the depth range between 1000-1500m. These data were available in the supplementary table, and my original regression calculations. I will now include them in the new Fig. 4 along with all of the data requested by the reviewer.**

*- Fig. 5d is a key figure and is much too small when printed. Moreover the DSi concentration is missing. I suggest to restrict Fig. 5 to the current panels a, b, c and to add a fig. 6 with current panel 5d + a panel with DSi concentration. Alternatively, Fig. 5 could cover a full A4 page and not just less than half of it.* **I agree with the reviewer that we could split Fig. 5d off as a figure of its own. I will include a new Fig. 5abc and Fig. 6.**

*- Could the authors provide a table with d30Si and DSi end-members of water masses C2 as calculated from their isotopic data and the contribution based on OMP from Garcia-Ibanez et al. (2017)? This would be very useful.* **Yes, we can do this, we can include it as a supplementary table.**

*- Supplementary Table S1: provide in the Table caption the definition of Si\* = DSi – NO3.* **We will make the changes recommended by the reviewer.**

---

## Author Response (AR1)

We would again like to thank Damien Cardinal and Patricia Grasse for their thorough reviews of our manuscript and their helpful comments. Here we present a point by point reply to each review separately, alongside the changes made to the manuscript. Note that the italicized text represents the comments made by the reviewers and the non-italicized/bold text is our response. Also, any changes made to the manuscript cite the page and line numbers from the modified "track changes" manuscript.

Cardinal Review
*P4 L2 vs. Table 1: in the text neb flow rate is 100 uL/min while it is 60 uL/min in Table 1. Homogenise.* **This has been changed in Table 1.**

*- Fig. 4 and in the text associated. 1) In this figure, the authors compare their GEOVIDE data with the two previous studies in the North Atlantic of Brzezinski & Jones (2015) and de Souza et al. (2012). Since Brzezinski & Jones chose to correct the offset between their data and the ones of de Souza et al. by +- 0.11 pmil, I suggest the authors here clearly mention that they always use the non-corrected data (which I believe is the right way to proceed) to avoid confusion with corrected data discussed in Brzezinski & Jones. 2) Important. Provide error bars of the three slopes and intercepts. Variability of GEOVIDE dataset seems higher. This should be checked and discussed. It is particularly needed given the offset found between the three data set that remains unsolved.* **We have taken into consideration the recommendations made by the reviewer and have made the following changes:**

**(1) The following text was modified on Page 7 Lines 16-22: "Our data agree with the systematics of these two studies, with each un-corrected dataset exhibiting similar linear regressions, except for the value of the y-intercepts, and the slope of the current data being slightly exaggerated relative to the other data-sets (see Fig. 3). Although the previously published data for the North Atlantic Ocean have nearly-identical linear regressions, an offset of +0.11 ‰, relative to the de Souza et al., (2012b) data was observed and discussed by Brzezinski and Jones (2015), concluding that an analytical bias existed."**

**(2) We modified the following text on page 8 Line 1: 'Factoring out the offset in absolute $\delta^{30}Si_{DSi}$ values and the greater variability in our data (see Fig. 3 for details).**

**(3) The figure caption for Fig. 3 (was Fig. 4) now includes the following information regarding the statistics associated with the linear regressions: "The statistics associated with each linear regression are as follows: de Souza et al. (2012b) – standard deviation of the slope (SDs) = 0.31, standard deviation of the y (SDy) = 0.06, standard deviation of the intercept (SDi) = 0.02, $R^2$= 0.86, n=58; Brzezinski and Jones (2015) - SDs = 0.32, SDy = 0.07, SDi = 0.02, $R^2$= 0.83, n=83; these results – SDs = 1.1, SDy = 0.17, SDi = 0.08, $R^2$= 0.64, n=29." on Page 21 Lines 18-22.**

**(4) In addition, I noticed that one data point was missing from the supplementary table (although had been included in Fig.3) and that several data points were missing from Fig. 3**

for the depth range between 1000-1500m. These data were available in the supplementary table, but not in the original regression calculations. This is why the slope has become slightly more positive in the new Fig. 3 (was Fig. 4)

*- Fig. 5d is a key figure and is much too small when printed. Moreover the DSi concentration is missing. I suggest to restrict Fig. 5 to the current panels a, b, c and to add a fig. 6 with current panel 5d + a panel with DSi concentration. Alternatively, Fig. 5 could cover a full A4 page and not just less than half of it.* **We have split Figure 5abc (now Fig. 4) and have made a new Fig. 5d (now Fig. 5b) and included silicic acid data (Fig. 5a) alongside Fig 5b. Both (new) Figures 4 and 5 are also plotted against longitude.**

*- Could the authors provide a table with d30Si and DSi end-members of water masses C2 as calculated from their isotopic data and the contribution based on OMP from Garcia-Ibanez et al. (2017)? This would be very useful.* **We have provided two supplementary tables (Table S3 and S4) with relevant information regarding DSi end members based on the OMP from** García-Ibáñez **et al., 2018. Table S3 provides all of the details regarding the DSi end members for the different water masses. Table S4 provides all of the d30Si values presented in this manuscript alongside the relevant information for each water mass at each sampling location. In addition, previously published values for AABW, LSW, DSOW, and AW are presented in Fig. 3 (was Fig. 4). The text referring to the endmembers in the figure caption is on page 21 Lines 22-24 and reads as: 'The stars presented in this figure represent the end-members for the AABW, LSW, and DSOW previously published by de Souza et al. (2012b – grey stars), the Arctic Water (AW) previously published by Brzezinksi and Jones (2015 – the black star), and an unknown end-member (blue star)."**

*- Supplementary Table S1:  provide in the Table caption the definition of Si\* = DSi – NO3.* **This definition was added to the supplementary Table S1 caption. In addition, we defined DSi as dissolved silicon in the caption for Table S1 and we added one data point that did not appear in the Table in the previous submission, although it did appear in older version of Fig. 5.**

Grasse Review

*P1 L12: I think the information that low DSI samples could not be measured does not necessarily need to be in the Abstract.   I think it would be more important that this information is mentioned in the methods or results part.* **We have removed the text in the abstract (Page 1 Lines 11-12) and have added the text to the results section (Page 6 Line 27-28).**

*P3 L20: Mesh size of the AG 1 X8 resin?* **The text "(100-200 mesh size)" was added to the manuscript (Page 3 Line 20).**

*P5 L15: It seems the manuscript Garcia-Inánez et al. is already accepted.* **This was not the case at the time of submission and has now been changed (Page 5 Line 15, Page 10 Line 7, Page 16 Line 10-12, Page 21 Line 7).**

*P6 L5: Please include one sentence about the DSI concentrations in the upper 500 m and that the samples could not be analyzed.* **Text has been added on Page 3 Lines 5-7: 'Dissolved silicon (DSi) concentrations were measured throughout the water column (Fig. 2, Supplementary Table S1), but only the samples collected below 500 m are discussed since no $\delta^{30}Si_{DSi}$ samples were collected from within the upper water column.'**

*P6 L18: Please give the exact values of the lowest δ30dSi (0.95 ‰ to 0.98 ‰. I think these very low δ30dSi values are actually quite interesting and need some more attention. See my comments below.* **We have added text on Page 6 Line 20: "0.95 ‰ and 0.98 ‰, respectively"**

*P7 L13: I think it would be helpful to modify figure 4. First of all, you should make your data more visible (e.g., bring your data to the front, use a light color for the already published data). You could try to group your data.e.g., only use open ocean stations vs. stations close to landmasses. Colorcode the stations or samples that are characterized by specific water masses. It would also be helpful to add water mass end members, e.g., AABW, which brings a light source from the south 1.2 ‰ (0.01 DSi; Souza et al. 2012). That could show additional processes that influence your deep-water masses e.g. at St. 1 and St. 13. Generally, I think it is interesting, that you see such light δ30dSi values and it should be discussed in more detail. According to your intercalibration with de Souza et al. (Fig. 6) and your results from the intercalibration study Grasse et al. (2017) your δ 30dSi data agrees very well within error (0.1 ‰ 2sd). Therefore, a water sample of 1 ‰ together with slightly higher DSi compared to de Souza et al., might indicate that further emineralization influences the δ30dSi composition. Such low (or even lower, 0.6 ‰ values are ypically associated with much higher DSi of 130 to 150 micromol in the Pacific and (Reynolds et al.2006, de Souza et al., 2012, Grasse et al., 2013) at DSi concentrations (even though I know that some people doubt some of the δ 30dSi deep water values in the North Pacific). However, Grasse et al. 2016 observed δ30dSi values of 1.1 ‰ in bottom water of the Peruvian shelf (â Li j40 micromol), which were influenced by pore waters from the sediment and remineralization at the sediment-seawater interface (Ehlert et al., 2016). Not necessary an effect you observe, but if not dissolution at the seawater-sediment interface or in the water column influences your δ 30dSi, you could also have admixture with a distinct water mass that brings in a very light δ30dSi signature (e.g., a water masses from Iceland? I am not so familiar with the water mass circulation in the Atlantic, but it seems that the NEADW can pick up its signature here?). Additionally, the circulation is quite sluggish, or? Therefore, you can have a trapping effect? I do not want, that you go too much into detail into the Pacific seawater δ 30dSi distribution and I also see that some of the values are identical within error, but I would like to have a better explanation why not all of your data does fall on the line for DSi versus δ 30dSi.* **We have taken into consideration the recommendations made by the reviewer and have made the following changes:**

**(1) The suggested changes (colour coding, end-members, etc.) made by the reviewer for Fig. 4**

(now Fig. 3) have been made. See Fig. 3 for modifications. In addition, after I received the reviews, I noticed that one data point was missing from the supplementary table (although had been included in Fig.4 – now Fig. 3) and that several data points were missing from Fig. 4 for the depth range between 1000-1500m. These data were available in the supplementary table, but I will now include them in the new Fig. 4 (Fig. 3). The lack of these data did modify the linear regression for this study, and the appropriate statistics were applied and discussed (see Cardinal Review regarding Fig. 4 (now Fig. 3) for details on the changes made to the manuscript).

**(2) The text referring to the endmembers in the figure caption is on page 21 Lines 22-25 and reads as: 'The stars presented in this figure represent the end-members for the circumpolar deep water (CDW), LSW, and DSOW previously published by de Souza et al. (2012b – grey stars), the Arctic Water (AW) previously published by Brzezinksi and Jones (2015 – the black star), and an unknown end-member (blue star)."**

**(3) The following text has been added to Page 10 Lines 16-17: "Interestingly, four of our deep samples from stations 1 and 13 have low $\delta^{30}Si_{DSi}$, perhaps indicative of another unknown source of isotopically light DSi (see Fig. 3). However, this still needs to be confirmed."**

*P7 L8: Please give the values (low, high) for the study by de Souza et al.* **These changes have been added to Page 7 Lines 12-13: "from high values (>2.0 ‰) in the Si-poor waters that contribute to NADW to low values (1.2 ‰).**

*P7 L25: please mention here (or at least above) the absolute $\delta30dSi$ values from the study of de Souza et al. for comparison with your values. The range can be similar, but that does not necessarily mean, that the $\delta30dSi$ are identical.* **This was addressed in the question above.**

*P10 L8: Please also explain, why the uppermost sample at station 26 has such high $\delta30dSi$.* **We have added the following text on Page 10 Line 23-24 – "At a depth of 500 m the $\delta^{30}Si_{DSi}$ value (+2.85 ‰) is our most elevated isotopic composition, which may be influenced by the SAIW, but this is difficult to argue since this sample site is the only partial sample from this water mass."**

*P11 L5: Please mention the stations you are talking about. High $\delta30dSi$? Value? What values?* **We have added the following text on Page 11 Line 19 – "..(e.g. STN 77, 502 m).."**

*P11 L10: What are the $\delta 30dSi$ values in the Labrador Sea? Please make clear that it is subducted surface water.* **We have added the following text on Page 11 Lines 24-25 – "(stations 64, 69, 77), which consists primarily of subducted surface water."**

*P11 L24: Please give me the station number and depth that makes it much easier to follow and understand your discussion.* **We have added the following text on Page 12 Line 6 – "..predominantly DSOW (STN 44, 2900 m).."**

*P11 L25 Doesn't NEADW has high DSi? Here I am getting confused, isn't the NEADW influencing the eastern deep waters?  At least according to Fig 4.  in Garcia-Ibanez et al.?  Please check the Garcia-Ibanez paper for water masses; it seems that there are some discrepancies, most likely as a result of the review process of the manuscript*. **As mentioned in the response to reviewers (starting page 5 Line 31), NEADW recirculates in the West European Basin and mixes with the surrounding waters, including the Antarctic Bottom Water (AABW) (van Aken and Becker, 1996), resulting in the formation of LDW. The concentration of DSi in the NEADW is.  NEADW is the dominant water mass for the deep Eastern area of OVIDE. As answered in the previous comment, NEADW is derived from ISOW so it is consistent to have similar d30Si in both ISOW and NEADW.**

*Fig.2 I do not think that the Figures has to be in the Paper. In my opinion, it is enough to mention in the text, that all samples fall on the mass-dependent fractionation line.* **We have made the changes recommended by the reviewer and have removed Fig. 2.**

*Fig4:  Can you please adjust the y-scale from 0.5 ‰ to 2 ‰.  Please add the studies indicated by different color directly to the legend.  Would be good to modify the figure (see comments above) C4* **These changes have been made (see Fig. 3).**

*Fig.  5:  It is quite tricky to distinguish the colors of different water mass types.  You could only name the dominant water mass in the figure. Similar to Garcia-Ibanez et al. (Figure 4). Can you replace section distance with longitude?* **These changes have been made (see Fig. 5).**

The following is a list of the relevant changes made in the manuscript described by page number. Note that all of the major changes are also indicated in the point by point response to reviewer comments.

The original **Figure 2** was removed and therefore the **figure captions for Figures 2-5 were changed** (see page 21 and 22 for modifications to figure captions)

**Figure 4 (now Figure 3) was heavily modified.** This figure now has (1) more data (data was available in the original supplementary table), (2) a modified linear regression for the current study's data (discussion was also modified – see below), (3) end-members for different water masses, (4) colour-coded data by station number for this study, and (5) relevant information (e.g. citation and linear regressions) for the two published data sets shown alongside the current data. The Figure caption (page 21) for this figure was modified to account for these changes and text was added for the discussion (page 7, see below) of these changes.

**Figure 5abc (now Figure 4)** was modified by splitting panels a, b, and c from panel d. Also, the x-axis is now represented by longittude instead of section distance.

**Figure 5d (now Figure 5b)** was split from Fig 5abc and a new panel (a) was added with silicic acid concentration data.

**Page 1**
**Inserted** "(GEOTRACES GA-01)"| in the Abstract and in the Title
**Deleted** "Near-surface water δ30SiDSi could not be evaluated due to the very low dissolved silicon (DSi) concentrations (< 5□M). However, v"

**Page 2**
**Inserted** "(GEOTRACES GA-01) in the introduction

**Page 3**
**Inserted** "100-200mesh size"

**Page 4**
**Inserted** "(d29SiNBS28 = 0.52 × d30SiNBS28)
**Deleted** "Fig. 2"

**Page 6**
**Inserted** "( on (DSi) concentrations were measured throughout the water column (Fig. 2a, Supplementary Table S1), but only the samples collected below 500 m are discussed since no $\delta^{30}Si_{DSi}$ samples were collected from within the upper water column."
**Inserted** "0.95 ‰ and 0.98 ‰, respectively"
**Inserted** "Near-surface water $\delta^{30}Si_{DSi}$ could not be evaluated due to the very low dissolved silicon (DSi) concentrations (< 5"μM)

**Page 7**
**Inserted** "(>2.0 ‰)"
**Inserted** "(1.2 ‰)"
**Changed the third paragraph to:** Our data agree with the systematics of these two studies, with each un-corrected dataset exhibiting similar linear regressions, except for the value of the y-intercepts, and the slope of the current data being slightly exaggerated relative to the other data-sets (see Fig. 3).

Although the previously published data for the North Atlantic Ocean have nearly-identical linear regressions, an offset of +0.11 ‰, relative to the de Souza et al., (2012b) data was observed and discussed by Brzezinski and Jones (2015), concluding that an analytical bias existed. Such offsets of order ±0.2 ‰ have been recognized to exist between seawater $\delta^{30}Si$ data produced in different laboratories (Grasse et al. 2017); their origin remains unclear, although they may have to do with differences in sample processing and chemical purification. The offset of the new data to that of de Souza et al. (2012b), produced at ETH Zurich, is somewhat surprising given the good agreement in $\delta^{30}Si_{DSi}$ for 6 seawater samples analyzed both at Plouzané and Zurich, but a small offset to lower $\delta^{30}Si_{DSi}$ at Plouzané is consistent with the offset (0.1 ‰) in these two laboratories' mean $\delta^{30}Si_{DSi}$ values for the seawater reference Aloha-1000 (Grasse et al., 2017). Whilst not ideal for the determination of the absolute $\delta^{30}Si_{DSi}$ value for each basin, the existence of such interlaboratory offsets does not impair our ability to analyze the distribution of $\delta^{30}Si_{DSi}$ along the GEOVIDE transect, with the systematics of our data exhibiting similar behaviour to previously-published studies (Fig. 3).

**Page 7**
**Inserted** "and the greater variability in our data (see Fig. 3 for details)."

**Page 10**
**Replaced** STN with stations
**Inserted** "Interestingly, four of our deep samples from stations 1 and 13 have low ☐$^{30}Si_{DSi}$, perhaps indicative of another unknown source of isotopically light DSi (see Fig. 3). However, this still needs to be confirmed."
**Inserted** "a depth of 500 m, the $\delta^{30}Si_{DSi}$ value (+2.85 ‰) is our most elevated isotopic composition, which may be influenced by the SAIW, but this is difficult to argue since this sample site is the only partial sample from this water mass, At"

**Page 11**
**Inserted** "(e.g. STN 77, 502 m)"
**Inserted** "and Fig. 5"
**Inserted** (stations 64, 69, 77), which consists primarily of subducted surface water"

**Page 12**
**Inserted** "predominantly" and "(STN 44, 2900 m)"

**Page 16**
**Deleted** " García-Ibáñez, M.I., Pérez, F.F., Lherminier, P., Zunino, P., Tréguer, P., in review: Water mass distributions and transports for the 2014 GEOVIDE cruise in the North Atlantic, Biogeosciences Discuss., doi:10.5194/bg-2017-355, 2017.
**Inserted** "García-Ibáñez, M. I., Pérez, F. F., Lherminier, P., Zunino, P., Mercier, H., and Tréguer, P.: Water mass distributions and transports for the 2014 GEOVIDE cruise in the North Atlantic, Biogeosciences, 15, 2075-2090, https://doi.org/10.5194/bg-15-2075-2018, 2018."

**Page 22**
**Acknowledgements -** a few changes were made here as well.

**Supplementary Tables**
**Two additional supplementary tables were created** to support the data shown in Figures 3 and 4.

[revised manuscript text omitted]